# Comparisons of Extracellular Vesicles from Human Epidural Fat-Derived Mesenchymal Stem Cells and Fibroblast Cells

**DOI:** 10.3390/ijms22062889

**Published:** 2021-03-12

**Authors:** Soo-Eun Sung, Kyung-Ku Kang, Joo-Hee Choi, Si-Joon Lee, KilSoo Kim, Ju-Hyeon Lim, Seung Yun Yang, Seul-Ki Kim, Min-Soo Seo, Gun Woo Lee

**Affiliations:** 1Department of Laboratory Animal Center, Daegu-Gyeongbuk Medical Innovation Foundation (DGMIF), Daegu 41061, Korea; sesung@dgmif.re.kr (S.-E.S.); kangkk@dgmif.re.kr (K.-K.K.); cjh522@dgmif.re.kr (J.-H.C.); sjlee1013@dgmif.re.kr (S.-J.L.); or kskim728@knu.ac.kr (K.K.); 2Department of Biomaterials Science (BK21 Four Program), Life and Industry Convergence Institute, Pusan National University, Miryang 50463, Korea; syang@pusan.ac.kr; 3College of Veterinary Medicine, Kyungpook National University, 80 Daehakro, Buk-gu, Daegu 41566, Korea; 4New Drug Development Center, Osong Medical Innovation Foundation, Chungbuk 28160, Korea; sinistemcells@kbiohealth.kr; 5Department of Orthopedic Surgery, Yeungnam University Medical Center, Yeungnam University College of Medicine, 170 Hyonchung-ro, Namgu, Daegu 42415, Korea; 6Efficacy Evaluation Team, Food Science R&D Center, KolmarBNH CO., LTD, 61Heolleungro 8-gil, Seocho-gu, Seoul 06800, Korea; lovesshot@kolmarbnh.co.kr

**Keywords:** extracellular vesicle, mesenchymal stem cell, epidural fat, fibroblast, inflammation, cytokine, chemokine

## Abstract

Extracellular vesicles (EVs) are generated and secreted by cells into the circulatory system. Stem cell-derived EVs have a therapeutic effect similar to that of stem cells and are considered an alternative method for cell therapy. Accordingly, research on the characteristics of EVs is emerging. EVs were isolated from human epidural fat-derived mesenchymal stem cells (MSCs) and human fibroblast culture media by ultracentrifugation. The characterization of EVs involved the typical evaluation of cluster of differentiation (CD antigens) marker expression by fluorescence-activated cell sorting, size analysis with dynamic laser scattering, and morphology analysis with transmission electron microscopy. Lastly, the secreted levels of cytokines and chemokines in EVs were determined by a cytokine assay. The isolated EVs had a typical size of approximately 30–200 nm, and the surface proteins CD9 and CD81 were expressed on human epidural fat MSCs and human fibroblast cells. The secreted levels of cytokines and chemokines were compared between human epidural fat MSC-derived EVs and human fibroblast-derived EVs. Human epidural fat MSC-derived EVs showed anti-inflammatory effects and promoted macrophage polarization. In this study, we demonstrated for the first time that human epidural fat MSC-derived EVs exhibit inflammatory suppressive potency relative to human fibroblast-derived EVs, which may be useful for the treatment of inflammation-related diseases.

## 1. Introduction

Extracellular vesicles (EVs) are known to be generated in most cells through the intracellular vesicular pathway and have an approximate diameter of 30–200 nm with spherical and double-lipid membrane structures [1]. EVs can be classified into several types based on size and origin [2]. EVs can be isolated from many biofluids. These include saliva, breast milk, urine, blood, cerebrospinal fluid, and cell culture media [3,4]. EVs consist of various RNAs, microRNAs (miRNAs), lipids, and proteins and play pivotal roles in cell-to-cell communication [5]. The components of EVs can stably exist, and EVs can pass through the blood–brain barrier on both sides [5]. Studies using EVs report the possibility of using genetic and protein information contained within EVs as diagnostic tools for various diseases, such as cardiovascular disease, inflammation, and cancer [6,7,8]. In addition, studies on therapeutic approaches and characterization using stem cell-derived EVs have been reported [9]. Therefore, it is necessary to confirm and analyze the characteristics of isolated EVs. Mesenchymal stem cells (MSCs) have therapeutic effects on various diseases. These include inflammatory diseases, and related studies have been continuously reported [10]. MSCs can be isolated from diverse tissues, such as umbilical cord blood, bone marrow, muscle, and adipose tissue [10,11,12]. In particular, various studies have been reported on the characteristics and therapeutic effects of adipose tissue-derived MSCs [13,14,15]. MSCs have also been isolated from epidural adipose tissue with reports of therapeutic effects [11,16]. Although MSCs have a variety of effects on tissue repair and meaningful therapeutic effects in the treatment of various diseases, due to several limitations of MSCs, new therapeutic strategies are required to treat additional disease and cells, [17]. As an alternative, recent studies have investigated the therapeutic effects associated with stem cell-derived EVs [18,19,20,21]. It has been possible to extract EVs from epidural adipose-derived mesenchymal stem cells, and such EVs have been confirmed to exhibit anti-inflammatory effects in a spinal cord injury animal model [21]. The epidural fat surrounds the dura matter of the spinal cord or nerve root [22,23]. However, the role and function of epidural fat are not clearly understood. In addition, studies on the characteristics and functions of epidural fat MSC-derived EVs have not been reported.

In the current study, we isolated and characterized human epidural fat MSC-derived EVs and human dermal fibroblast-derived EVs from cell culture media. To compare the characteristics of EVs isolated from epidural fat, MSCs and dermal fibroblast, the secretory quantification of cytokines and growth factors were evaluated. With this information, we sought to confirm and analyze whether EVs derived from epidural fat MSCs exhibit therapeutic potency in inflammatory responses.

## 2. Results

### 2.1. Isolation and Characterization of Human Epidural Fat MSCs

Human epidural fat MSCs were isolated from epidural fat tissue obtained during spine surgery. Human epidural fat MSCs exhibit a spindle morphology that resembles that of human dermal fibroblasts (Figure 1A). All epidural fat MSCs isolated from individuals exhibited similar morphologies and characteristics. The epidural fat MSCs and dermal fibroblasts were attached to the bottom of the culture plates. To identify stem cell characteristics, fluorescence-activated cell sorting (FACS) analysis was conducted using specific or non-specific markers in stem cells. CD14, CD34, and CD45 were not expressed in the isolated epidural fat MSCs. However, CD73, CD90, and CD105, which are stem cell markers, were expressed in the epidural fat MSCs (Figure 1B). These results indicated that the isolated cells from human epidural fat exhibited MSC characteristics.

### 2.2. EVs from Human Dermal Fibroblasts and Human Epidural Fat MSCs

To harvest mass amounts of highly purified EVs from the cells, tangential flow filtration (TFF) was performed. Cultured conditioned media included many EVs. These included exosomes, microparticles, and apoptotic bodies. Among them, exosomes are critical particles because they contain nucleic acids, proteins, receptors, and major histocompatibility complex molecules [24]. To purify a high yield of exosomes, a modified protocol involving centrifugation or a TFF system was conducted. This was followed by optimizing control for exosome isolation. Most reported methods for exosome isolation involve only ultracentrifugation. However, we attempted the following process for isolating EVs: first, cell debris were removed from the cell culture medium by centrifugation. Second, the TFF system was used to filter molecules smaller than the size of the exosomes. Finally, microparticle-sized EVs were eliminated by centrifugation. Isolated EVs were visualized by transmission electron microscopy (TEM). We confirmed that the isolated EVs contained thick and dark lipid bilayer membranes with a size of approximately 100 nm (Figure 2A). Figure 2A showed the size of single exosome and the morphology of the membrane in detail. Exosome membranes displayed tetraspanin proteins, such as CD63 and CD81. Thus, the tetraspanin proteins were analyzed by flow cytometry. Dermal fibroblast-derived EVs and epidural fat MSC-derived EVs were both ultimately found to show the tetraspanins CD63 and CD81 (Figure 2B). Nanoparticle tracking analysis (NTA) was used to determine the size distribution and concentration of the isolated EVs (Figure 2C). Using NTA, dermal fibroblast-derived EVs were isolated at a density of 1.85 × 10^9^ particles/mL, and epidural fat MSC-derived EVs were isolated at a density of 2.35 × 10^9^ particles/mL. The total volume was approximately 5 mL. We harvested a minimum of 9.25 × 10^9^ particles and a maximum of 11.75 × 10^9^ particles from dermal fibroblasts and epidural fat MSCs, respectively. The dermal fibroblast-derived EV size was 180.4 ± 4.8 nm. Conversely, the size of the epidural fat MSC-derived EV was 176.9 ± 1.6 nm. Hence, the dermal fibroblast-derived EVs and epidural fat MSC-derived EVs exhibited similar sizes. However, the generating ability of EVs was better for epidural fat MSCs than for dermal fibroblasts.

### 2.3. Pro-Inflammatory Molecule Levels in Human Epidural Fat MSC-Derived EVs

MSCs and dermal fibroblasts exhibited similar proliferation and differentiation capacities, as well as cell surface markers. However, a difference emerged in the immunomodulation based on origin [25]. Since MSCs reduce inflammation by paracrine effects, we hypothesized that EVs from epidural fat MSCs secreted low levels of pro-inflammatory factors, such as cytokines and chemokines. Cytokine assays were performed to analyze pro-inflammatory and anti-inflammatory cytokines. Some cytokine and small molecule expression levels were altered in epidural fat MSC-derived EVs compared to those of dermal fibroblast-derived EVs (Figure 3). Secreted levels of GM-CSF, MIP-1 α, MMP-9, IL-5, GRO, and VEGF decreased in epidural fat MSC-derived EVs. In contrast, IL-4, IL-10, and IL-13 cytokines tended to increase in EVs from epidural fat MSCs. Although IL-4 and IL-10 cytokine levels were not significantly different, that of the anti-inflammatory factor IL-13 was significantly increased. Hence, epidural fat MSC-derived EVs showed low levels of some pro-inflammatory cytokines and high levels of the anti-inflammatory factor IL-13, in comparison with those of dermal fibroblast-derived EVs.

### 2.4. Anti-Inflammatory Effects of Human Epidural Fat MSC-Derived EVs

To confirm whether epidural fat MSC-derived EVs induced anti-inflammatory effects, the THP-1 human monocytic leukemia cell line was used. THP-1 cells can differentiate into macrophage-like cells via the protein kinase C activator phorbol 12-myristate 13-acetate (PMA) [26]. THP-1 cells cultured without PMA exhibited sufficient proliferation. PMA was added to THP-1 cells that were grown in culture plates. THP-1 cells that were treated with PMA stopped proliferating, attached to the bottom of culture dishes, and subsequently differentiated into macrophage-like cells (Figure 4A). Lipopolysaccharides (LPS) compose the outer membrane of Gram-negative bacteria and are considered among the best characterized pathogen-associated molecular patterns that interact with CD14/TLR4 and activate intracellular signals [27]. When the THP-1-derived macrophages were treated with LPS, pro-inflammatory factors and arachidonic acid products, such as PGE_2_ and PGF_2α_, were released [28]. To detect LPS-induced inflammation and cytokine expression changes, LPS stimulation was attempted. EV- to LPS-stimulated THP-1-derived macrophages were also treated to determine the anti-inflammatory effect of EVs on the macrophages (Figure 4A). After LPS stimulation, TNFα, IL6, and IL10 of the conditioned media were quantified by a cytokine array and ELISA (Figure 4B). LPS significantly induced TNFα (973.6 ± 0.69 pg/mL) in THP-1-derived macrophages. Conversely, the non-LPS-treated group and EV-treated group did not differ in terms of TNFα production. Dermal fibroblast, and epidural fat MSC-derived EVs treated along with LPS, exhibited significantly decreased TNFα production at values of 68.55 ± 49.92 pg/mL and 53.64 ± 33.20 pg/mL, respectively. The cytokine IL6 was not produced in the non-LPS-treated group. However, IL6 production increased in the LPS-treated group (1489.39 ± 121.92 pg/mL). Moreover, LPS-treated dermal fibroblast-derived EVs increased IL6 production levels (1321.79 ± 203.60 pg/mL). However, IL6 production was significantly blocked when LPS and epidural fat MSC-derived EVs were simultaneously treated (167.78 ± 7.69 pg/mL). When LPS and epidural fat MSC-derived EVs were simultaneously treated, IL-10 production was determined to be at the value of 406.96 ± 67.94 pg/mL. This is an increased value compared to treatment with only LPS (77.49 ± 15.80 pg/mL), and treatment with both LPS and dermal fibroblast-EVs (133.23 ± 15.80 pg/mL). These results demonstrate that EVs decreased TNF-α production in THP-1-derived macrophages. In particular, unlike dermal fibroblast-derived EVs, epidural fat MSC-derived EVs showed decreased pro-inflammatory factor IL-6 production. In addition, the production of IL-10, a known anti-inflammatory factor, was increased in epidural fat MSC-derived EVs. Thus, treatment with epidural fat MSC-derived EVs inhibited LPS-induced inflammation in THP-1 macrophages compared to that with dermal fibroblast-derived EVs.

## 3. Discussion

MSCs are known to play a pivotal role in the tissue regeneration and immunomodulation processes. Soluble factors are secreted from stem cells, affecting the recovery of damaged tissues and organs. Moreover, the effects of stem cell paracrine secretion have been reported. MSCs secrete growth factors, chemokines, and cytokines to modulate actions on adjacent cells. These secreted factors are involved in increasing angiogenesis, reducing apoptosis and fibrosis, and regulating the immune response [29]. Therefore, MSCs promote the regeneration of damaged cells through paracrine secretion. This, in turn, accelerates cell recovery. The therapeutic effect of MSCs has been reported in relation to immune diseases, neurological diseases, liver diseases, kidney diseases, heart diseases, and lung diseases [30,31].

EVs are a major component of the paracrine factors secreted by MSCs. MSC-derived EVs have been reported to have anti-inflammatory effects in various disease models [9]. MSC-derived EVs exhibit immunosuppressive functions by reducing T and B lymphocyte proliferation and inducing Treg cell populations. In addition, EVs have the effect of lowering the expression of TNF-α and increasing the expression of IL-10 by affecting macrophage maturation [9,32,33,34]. MSC-derived EVs are also able to overcome the limitations of conventional MSC-based therapy. For example, quality control is among the major issues in terms of “live cell materials” when treated with MSCs. Isolation, culture conditions, storage methods, and administration of MSCs have a significant impact on cell efficacy and survival and could also be problematic in terms of reproducibility and their high cost [35,36]. However, storage and transport with cell-free materials such as EVs are cheaper and more efficient than with MSCs. Since EVs are cell free, they are more stable than MSCs in immunogenicity and tumorigenicity [37,38,39]. These previous studies indicate that MSC-derived EVs are a suitable source for disease research due to their potential anti-inflammatory effects.

Adipose tissue has been shown to represent a rich source of MSCs, and studies have shown that EVs derived from human adipose tissue MSCs can be used to ameliorate tissue damage in various disease models [13,14,15]. Epidural fat tissue is inevitably exposed during spinal surgery and spontaneously diminishes after spinal surgery [23]. Recently, the need for research on the characteristics and functions of epidural fat tissue has emerged. Studies have reported that MSCs can also be isolated from epidural fat tissue, and that the isolated MSCs exhibited therapeutic effects [11,16,21,40,41]. However, the specific characteristics and functions of human epidural fat MSCs and human epidural fat MSC-derived EVs are not yet clear. Therefore, it is necessary to confirm the characteristics of epidural fat MSC-derived EVs. As such, our study was conducted to investigate and compare the characteristics of EVs derived from human epidural fat MSCs to those of EVs derived from human fibroblast cells.

EVs from human epidural fat MSCs and human fibroblast cells were successfully isolated using the ultracentrifuge method and validated as EVs using established characterization methods. In the inflammation mechanism, pro-inflammatory cytokines, such as tumor necrosis factor (TNF-α), interleukin-6 (IL-6), and IL-1β, act as major factors. These factors are produced by activated macrophages [35]. Conversely, anti-inflammatory cytokines are produced by regulatory T cells (Tregs) and helper T (Th2) cells that regulate inflammatory and immune responses [19,32,33,34,42,43]. Major anti-inflammatory cytokines include IL-4, IL-10, IL-13, and transforming growth factor β (TGF-β), which inhibit Th1 response and pro-inflammatory cytokine expression [44,45]. Previous studies have reported that MSC-derived EVs hold the capability of inducing macrophage polarization from M1 to M2. The M1 macrophage has the property of widely expressing pro-inflammatory cytokines and chemokines, such as IL-1, IL-12, IL-6, and TNF-α. Conversely, in M2 macrophages, the expression of anti-inflammatory factors, such as IL-10 and TFG-α, is induced by Th2 cytokines [46,47,48].

This study has some limitations. First, the therapeutic effect of human epidural fat MSC-derived EVs in specific disease animal models related to inflammation was not studied. Therefore, this must be evaluated in further studies. Second, the gene expression patterns of EVs from human epidural fat MSCs and fibroblast cells was not compared in the present study. Therefore, further studies on gene expression and other detailed investigations are needed. Finally, the obtained outcome is currently limited to in vitro circumstances. Nevertheless, from the outcomes of the present study, epidural fat tissue may possess some useful abilities in the anti-inflammatory pathway. Epidural fat MSC-derived EVs may have the potential to manage immune-related diseases, but further studies are needed to better define the function and role of epidural fat tissue. Our study has several strengths. To our knowledge, the present study is the first to demonstrate that human epidural fat MSC-derived EVs exhibit anti-inflammatory effects via lowering the expression levels of pro-inflammatory cytokines and chemokines, as well as raising the expression levels of anti-inflammatory cytokines. We are also the first to compare the anti-inflammatory and pro-inflammatory cytokines of EVs from human epidural fat MSCs and human fibroblast cells. The cytokine assay outcomes revealed that the epidural fat MSC-derived EVs possessed more anti-inflammatory and less pro-inflammatory cytokines than fibroblast cell-derived EVs. Based on the results of the present study, we suggest that human epidural fat MSC-derived EVs are crucial for therapeutic approaches to inflammation-related spine diseases. Moreover, these results suggest that such EVs can be used as cell-free alternatives to stem cell-based therapeutics.

## 4. Materials and Methods

### 4.1. Cell Isolation and Culture

This study was approved by the Institutional Review Board (IRB) and Ethical Committee of Yeungnam University Medical Center in Daegu, South Korea (IRB no. 2017-07-032). After approval, the epidural fat tissue was obtained from consenting patients (n = 5) during posterior decompression surgery of the lumbar spine performed at our center. Epidural fat MSCs were derived from epidural adipose tissue through several established steps. The extracted epidural fat was dipped in PBS; blood vessels were separated and rinsed in 70% EtOH to prevent contamination with other germs and washed with ice-cold PBS. The tissue was digested and incubated in 0.45 µm filtered collagenase type I 2 mg/mL (Gibco, Invitrogen, Carlsbad CA, USA, 17018029) at 37 °C for 30 min. All tissues were digested and filtered using a 70 µm strainer, and the solution was then centrifuged at 3000× *g* for 5 min. Human dermal fibroblasts were purchased from American Type Culture Collection (ATCC, Manassas, VA, USA). Human dermal fibroblasts and human epidural fat MSCs were maintained in low glucose-DMEM (Gibco, Carlsbad CA, USA) supplemented with 10% exosome-depleted FBS (Gibco), 100 U/mL penicillin, 100 µg/mL streptomycin (Gibco), 10 µg/mL recombinant human FGF-basic (Peprotech, Seoul, South Korea), 10 µg/mL recombinant human PDGF-BB (Peprotech), and 25 µg/mL plasmocin (InvivoGen, San Diego, CA, USA). Human THP-1 cells were purchased from ATCC. THP-1 human monocytic cells were cultivated in ATCC-formulated RPMI-1640 medium containing 10% FBS (Gibco) and 2-mercaptoethanol (Gibco) to a final concentration of 0.05 mM, 100 U/mL penicillin, and 100 µg/mL streptomycin. All cells were maintained at 37 °C and 5% CO_2_ in a humidified atmosphere.

### 4.2. EV Isolation

At passage numbers 5 to 8, human dermal fibroblasts and human epidural fat MSC culture media were collected every 48 h. The cultured medium from each source was stored at 4 °C and pooled until a total volume of 50 mL was obtained. The medium was then centrifuged at 300× *g* for 10 min, and cells debris was discarded. Next, the medium was concentrated and filtered using a TFF system (Pall Corporation, Minimate TFF™ system, Port Washington, New York, NY, USA). The feed flow was approximately 12.5 mL/min. The final volume of the epidural fat MSC-derived EVs was 5 mL. The nominated pore size of the TFF system filter membrane used was 10 nm (Pall Corporation, Minimate TFF capsule, 100K). After the TFF process, the solution was centrifuged at 2500× *g* for 25 min to remove microparticles. EV protein concentration was measured using a Pierce™ BCA assay kit (Thermo Fisher Scientific, Rockford, IL, USA).

### 4.3. Flow Cytometry

FACS was performed using BD FACSAria III (Becton, Dickinson and Company, Franklin Lakes, NJ, USA). To analyze the mesenchymal stem cell markers, 10^5^ cells were stained with antibodies, particularly PE-conjugated CD105 (Bio-Rad, Hercules, CA, USA, MCA1557,), PE-conjugated CD90 (BioLegend, San Diego, CA, USA, 555596), PE-conjugated CD73 (BioLegend, 344004), FITC-conjugated CD45 (BioLegend, 555482), FITC-conjugated CD34 (BioLegend, 343504), and PE-conjugated CD14 (Bio-Rad, MCA1568). To determine the EV-specific marker, isolated EVs were incubated with 4% *w*/*v* aldehyde/sulfate-latex beads (Thermo Fisher Scientific, Rockford, IL, U.S, A37304) at room temperature overnight on rotation. Bead-binding extracellular cells were centrifugated at 1800× *g* for 10 min and washed with 500 µL of PBS. The pellet was resuspended with 50 µL of PBS containing CD63 (BioLegend, 353003) and CD81 (BioLegend, 349505) at 4 °C for 1 h. All antibodies were conjugated with PE fluorescence dye. Samples were washed with 500 µL of PBS and centrifuged at 1800× *g* for 10 min. The pellet was resuspended with PBS. Gating of exosome-decorated beads approximately 4 µm in diameter was analyzed using BD FACSAria III (BD Biosciences, San Jose, CA, USA). At least 10,000 events were acquired on flow cytometry. Data analyses was performed using BD FACSDiva™ software version 6.1.3 (BD Biosciences, San Jose, CA, USA).

### 4.4. TEM

EVs were absorbed onto a formvar carbon-coated copper grid (Ted Pella Inc., Redding, CA, USA, 01800-F), fixed in 2% paraformaldehyde for 10 min, and then dried. Grids were visualized using a biotransmission electron microscope (Hitachi, Japan, HT7700).

### 4.5. NTA

NTA analyses were performed using a Nanosight NS300 (Malvern Panalytical, Worcestershire, UK) instrument, as advised by the manufacturer.

### 4.6. Cytokine Assay

Cytokines were detected using a sandwich ELISA-based quantitative array platform. Dermal fibroblast-derived EVs and epidural fat MSC-derived EVs were measured at a protein concentration of 1.3 mg/mL to analyze cytokine levels. Cytokine levels were measured using a commercial array kit according to the manufacturer’s protocol (RayBiotech, Peachtree Corners, GA, USA, QAH-CYT-1).

### 4.7. LPS and EV Treatment of THP-1 Cells

Human THP-1 cells were differentiated into macrophages with 100 nM PMA (Sigma-Aldrich, Saint Louis, MO, USA) for 24 h. Before LPS treatment, PMA from THP-1 cells was washed. Human dermal fibroblast-derived EVs and human epidural fat MSC-derived EVs (50 µg/mL) were added simultaneously with 1 µg/mL LPS (Sigma-Aldrich). As controls, THP-1-derived macrophages were cultured either without LPS and EVs or with 50 µg/mL of two cell-derived EVs.

### 4.8. Statistical Analysis

Statistical analyses were performed using Prism (GraphPad Software, San Diego, CA, USA). All data are presented as mean ± SD. The significance of the differences was analyzed using a Student’s *t*-test or a one-way ANOVA. A *p*-value less than 0.05 was considered a significant difference.

## Figures and Tables

**Figure 1 ijms-22-02889-f001:**
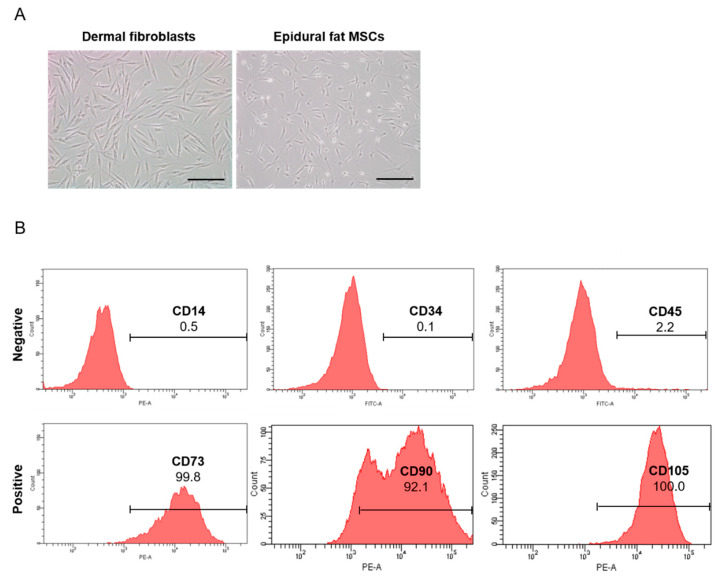
Characteristics of human epidural fat mesenchymal stem cells (MSCs). (**A**) Representative bright field microscopy images of human dermal fibroblasts and human epidural fat MSCs (original magnification 100x, scale bar = 25 µm). (**B**) Flow cytometric analysis showing expressed (CD73, CD90, and CD105) and non-expressed (CD14, CD34, and CD45) human epidural fat MSC. The horizontal axis is the fluorescence intensity and the vertical axis is the cell count.

**Figure 2 ijms-22-02889-f002:**
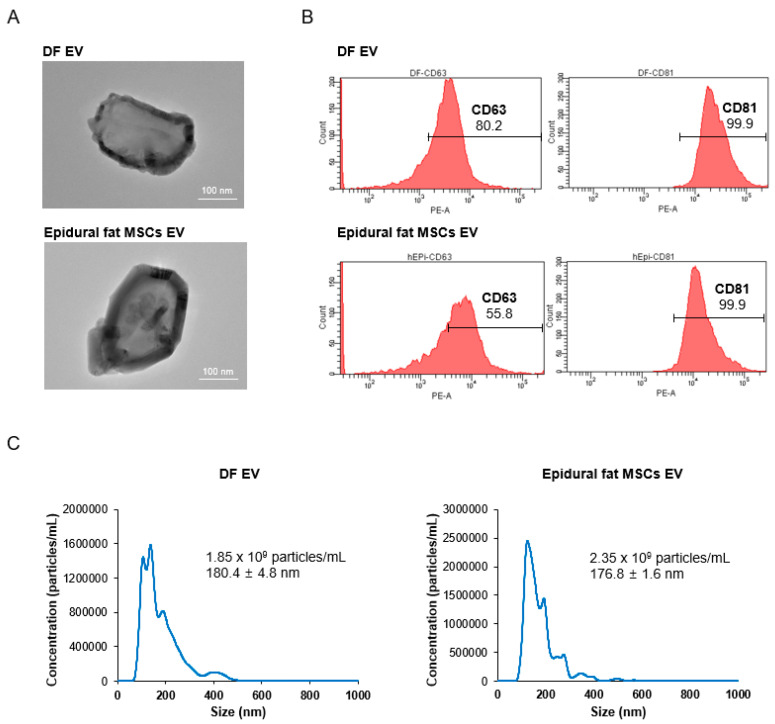
Identification of isolated human dermal fibroblast (DF)- and epidural fat mesenchymal stem cell (MSC)-derived extracellular vesicles. (**A**) Morphology of human dermal fibroblasts and human epidural fat MSCs shown by transmission electron microscopy (TEM). (**B**) Flow cytometric analysis showing the tetraspanin markers displayed on extracellular vesicles, such as CD63 and CD81. The horizontal axis is the fluorescence intensity, and the vertical axis is the cell count. (**C**) Size distribution and concentration of isolated extracellular vesicles by nanoparticle tracking analysis (NTA).

**Figure 3 ijms-22-02889-f003:**
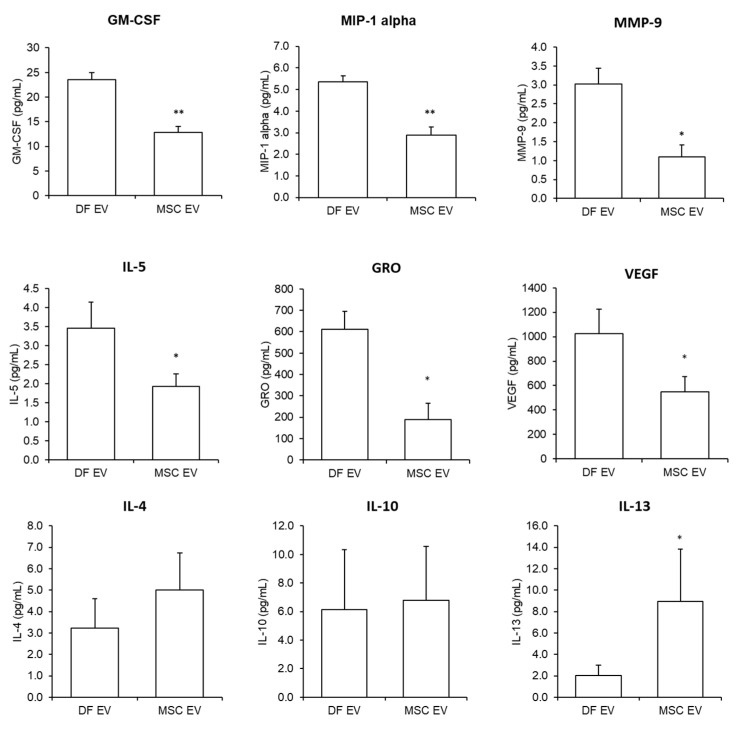
Cytokine and chemokine levels from human dermal fibroblasts and human epidural fat mesenchymal stem cell (MSC)-derived extracellular vesicles. GM-CSF, MIP-1 alpha, MMP-9, IL-5, GRO, and VEGF decreased in human epidural fat MSC-derived extracellular vesicles. IL-4, IL-10, and IL-13 increased in human epidural fat MSC-derived extracellular vesicles. (* *p*-value < 0.05, ** ≤ 0.001).

**Figure 4 ijms-22-02889-f004:**
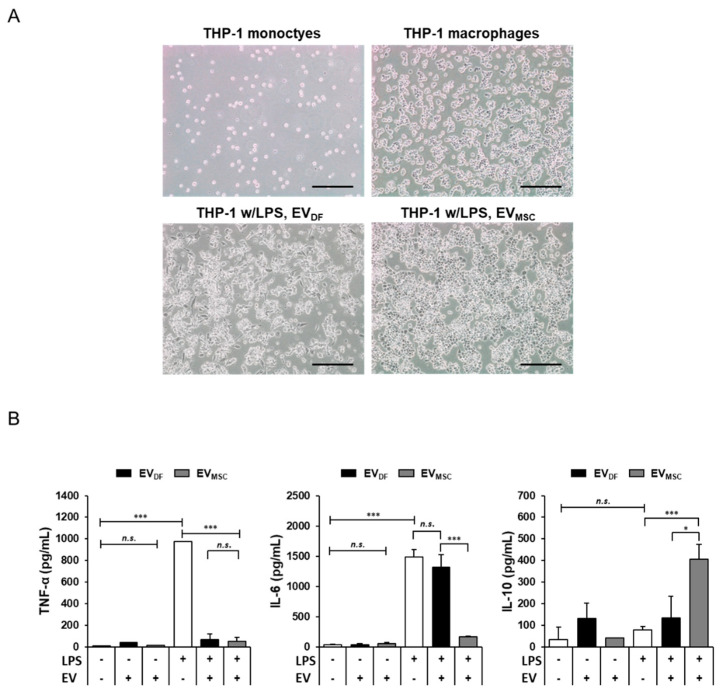
Anti-inflammatory effects of human dermal fibroblasts and human epidural fat MSC-derived extracellular vesicles on THP-1 macrophage cells. (**A**) Human monocyte THP-1 cells were treated with phorbol 12-myristate 13-acetate (PMA, 100 nM) and differentiated into macrophages. PMA-differentiated THP-1 cells were treated with lipopolysaccharides (LPS) and extracellular vesicles. (**B**) Quantification of TNF-α, IL-6, and IL-10 secretion levels by PMA-differentiated THP-1 cells either stimulated or untreated with LPS and/or extracellular vesicles (EVs) for 24 h. (* *p*-value < 0.05, *** < 0.0001, *n.s.* non-significant).

## Data Availability

The data that support the findings of this study are available from the corresponding author upon reasonable request.

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
