# Peer review of "Comparisons of Extracellular Vesicles from Human Epidural Fat-Derived Mesenchymal Stem Cells and Fibroblast Cells"

_ijms, 2021, doi:10.3390/ijms22062889_

Round 1

Reviewer 1 Report

Accepted in current form. 

Reviewer 2 Report

I can recommend the revised version of this paper for publication.

This manuscript is a resubmission of an earlier submission. The following is a list of the peer review reports and author responses from that submission.

Round 1

Reviewer 1 Report

COMMENTS

  1. Samples were obtained from decompresion surgery, it may happen that this bias all the results. MSC are activated by local or nearly located injuries and the secretome or the EVs reflect the response to the physiopathological situation.
  2. Fibroblasts EV are not the appropiate control
  3. EVs do not "express" but show or display. Only cells may express either membane markers or cytokines, etc
  4. page 6: EVs were treated with the LPS-stimulated THP-1... It seems was the opposite: EVs acted as antiinflammatory of the macrophage like THP-1...
  5. Discussion should be shortened and centered on the introduction and results

Reviewer 2 Report

This is an interesting study, it will give valuable information to the readers, this manuscript will provide basis for the in vivo applications of stem cells derived products to the scientific community. 

The isolation of EVs by centrifugation is described in details but insufficient information are given in the methods, which need to be improved. 

Reviewer 3 Report

The manuscript “Comparisons of Extracellular Vesicles from human Epidural Adipose-derived Mesenchymal Stem Cells and Fibroblast Cells” reports original results examining the properties of extracellular vesicles (EVs) derived from the cells of epidural fat tissue and skin fibroblasts. The main finding is that EVs from epidural fat mesenchymal stem cells can induce or be associated with anti-inflammatory responses. Although the results are well documented and presented, few clarifications and corrections are needed as listed below:

1) the abstract says that the size of EVs was approximately 100 nm while the results indicate approximately doubled diameter of 180 and 177 nm.

2) Figures 1A and 2B do not explain the numbers on the flow cytometry histograms.

3) Fig 2A: can you show images with multiple EVs?

4) source and method of isolation of human dermal fibroblast are required.

5) statistics: what was a post-hoc test used with one-way ANOVA?

6) overall the text is well-written, however several spots need editing: lines 82-83, 111, 119-120, 144-145, 190-191, 222-223, 229, 274, and Figure 4 typo (monocytes).